# Effect of Fluoride Content of Mouthwashes on the Metallic Ion Release in Different Orthodontics Archwires

**DOI:** 10.3390/ijerph20042780

**Published:** 2023-02-04

**Authors:** Francisco Pastor, Juan Carlos Rodriguez, José María Barrera, José Angel Delgado García-Menocal, Aritza Brizuela, Andreu Puigdollers, Eduardo Espinar, Javier Gil

**Affiliations:** 1Department of Ortodoncia, Facultad de Odontología, Universidad de Sevilla, Avicena s/n, 41009 Sevilla, Spain; 2Bioengineering Institute of Technology, Facultad de Medicina y Ciencias de la Salud, Universidad Internacional de Catalunya, Josep Trueta s/n, Sant Cugat del Vallés, 08195 Barcelona, Spain; 3Facultad de Odontología, Universidad Europea Miguel de Cervantes, C/del Padre Julio Chevalier 2, 47012 Valladolid, Spain; 4Department Ortodoncia, Facultad de Odontología, Universidad Internacional de Catalunya, Josep Trueta s/n, Sant Cugat del Vallés, 08195 Barcelona, Spain

**Keywords:** orthodontic arch wires, ion release, nickel, mouthwashes, fluoride, NitI, Ti-Mo, stainless steel

## Abstract

Metal ion release studies were carried out on three of the most commonly used orthodontic wires in the clinic: austenitic stainless steel, Ti-Mo, and superelastic NiTi, using three mouthwashes with different fluoride concentrations: 130, 200, and 380 ppm. Immersions were carried out in these mouthwashes at 37 °C for 1, 4, 7, and 14 days, and the ions released were determined by inductively coupled plasma-mass spectrometry (ICP-MS). All wires were observed by scanning electron microscopy (SEM). The results showed a moderate ion release in the stainless steel wires, with nickel and chromium values of 500 and 1000 ppb in the worst conditions for the wires: concentrations of 380 ppm fluoride and 14 days of immersion. However, in the Ti-Mo and NiTi alloys, an abrupt change in release was observed when the samples were immersed in 380 ppm fluoride concentrations. Titanium releases in Ti-Mo wires reached 200,000 ppb, creating numerous pits on the surface. Under the same conditions, the release of Ni and Ti ions from the superelastic wires also exceeded 220,000 ppb and 180,000 ppb, respectively. This release of ions causes variations in the chemical composition of the wires, causing the appearance of martensite plates in the austenitic matrix after 4 days of immersion. This fact causes it to lose its superelastic properties at a temperature of 37 °C. In the case of immersion in 380 ppm mouthwashes for more than 7 days, rich-nickel precipitates can be seen. These embrittle the wire and lose all tooth-correcting properties. It should be noted that the release of Ni ions can cause hypersensitivity in patients, particularly women. The results indicate that the use of mouthwashes with a high content of fluoride should not be recommended with orthodontic archwires.

## 1. Introduction

Orthodontic metals present good biocompatibility, mechanical properties, and corrosion resistance. Orthodontists can choose from a wide range of wires, mini-implants, or brackets made from stainless steel, Ni-Ti, Ni-Ti-Cu, Ti-Mo, and Cr-Co alloys. However, they can lose their protective oxide layer, leading to the release of metallic ions in the physiological medium. Ion release will be influenced by the chemical composition, the crystalline structure present (stainless steel is face-centered cubic, NiTi is austenite B2 body-centered cubic, martensite is monoclinic, and Ti-Mo is orthorhombic), and the bond energy. This release can be accelerated by mouthwashes and/or gels that contain fluoride [1,2,3]. The ions (especially nickel and chromium) can cause hypersensitivity, allergies, or modifications in cellular morphologies [4,5,6,7].

The Nickel’s harmful effects have been studied [8,9,10] because approximately 10% of the general population has hypersensitive reactions to this metal. This effect is ten times more common in women than in men [2]. Different alterations have been demonstrated, such as contact dermatitis, asthma, and cytotoxicity, among others. In addition, chromium ions are also cytotoxic [2], causing local and systemic reactions of hypersensitivity.

The orthodontic treatment can have periodontal implications, such as demineralization of the dental enamel and the appearance of white spots or even caries. This fact provokes the use of mouthwashes containing fluoride, which prevents caries due to the formation of fluorapatite in the mineral content of the teeth, which is much more resistant to caries and has a beneficial action in the treatment of dental hypersensitivity [11,12,13]. This makes the use of gels and mouthwashes common during orthodontic treatment. However, the concentration of fluoride in aqueous or alcoholic solutions promotes the reaction of fluorides with the metals that form orthodontic wires, especially Ti, causing the release of metal ions into the environment [2,14,15,16]. Mouthwashes have different concentrations of sodium fluoride, which in physiological media forms HF because all sodium salts are soluble in aqueous media.

Some studies have shown that fluoride solutions of varying pH can modify mechanical properties such as friction coefficients, the superelastic behavior of NiTi archwires, and corrosion behavior [17,18,19,20,21,22,23,24]. The fluoride ions are very aggressive with the oxide film, especially titanium dioxide, producing electrochemical corrosion. This corrosion produces roughness on the surface, increasing the friction coefficient between the archwire and the bracket. This roughness decreases bracket-wire sliding and therefore negatively affects tooth movement [25].

Ion release was determined with the three types of wires most commonly used in orthodontics: austenitic stainless steel, Ti-Mo, an1d NiTi. The arch wires were immersed in three concentrations of fluoride solutions widely used in commercial mouthwashes. The tests were carried out at 37 °C. The study of ion releases from orthodontic wires is not very common, since most publications refer to brackets, which will be in the mouth for a longer period. Also, a fluoride solution is usually used, which is generally 130 ppm. However, nowadays there are mouthwashes with a higher concentration of fluoride to achieve greater efficacy. In this work, we have used inductively coupled plasma-mass spectrometry (ICP-MS) techniques, which are of high resolution. This work aims to respond to the request of the European dental health authorities regarding the biodegradation of orthodontic elements with respect to treatments with fluoride mouthwashes.

## 2. Materials and Methods

Sixty commercial archwires of various alloys were investigated. The chemical compositions are shown in Table 1. These percentage compositions were determined using the dispersive energy of X-rays.

The wires were placed in the mouthwashes with the same chemical composition except for the sodium fluoride content at a constant temperature of 37 °C. The sodium fluoride contents can be seen in Table 2. Commercial mouthwashes can be classified as having low sodium fluoride content up to approximately 200 ppm and high sodium fluoride content with a high bactericidal capacity from 300 ppm on. To compare the behavior of different orthodontic archwires with the same surface, these studies were conducted. Five orthodontic wires were analyzed for each alloy and for each mouthwash solution, as well as for the control.

5 arch wires × 3 alloys (NiTi, TiMo, and stainless steel) × 4 concentrations (0, 180, 200, and 380 ppm) = 60 samples.

The ion release test was performed by immersing the archwires in 6 mL of the three different mouthwashes (Table 2) at 37 °C for 1, 4, 7, and 14 days. The immersion times were obtained from the conclusions of several health authorities and exposed at the International Orthodontic Conference [26,27,28,29], where orthodontic clinicians noted that in some cases, archwires and other devices were introduced at sleep times in mouthwashes with different concentrations of sodium fluoride [29,30]. Mouthwash must be used twice a day for about 90 s. It is recommended that the patient not eat or drink after the treatment and rinse, so that the mouthwash components remain present for an extended period of time. This is the reason it is difficult to determine the duration of contact between orthodontic archwires and mouthwashes. Different authors [30,31,32,33] assumed that the time that the mouthwash was present in the patient’s mouth for brackets was estimated at 45 days, and for archwires, it was about 14 days. Treatments could in some cases last as long as 25 days, and it was during these times that insight could be gained into the behavior of NiTi wires [33,34]. Ion-release quantification was carried out by inductively coupled plasma-mass spectrometry (ICP-MS) using Perkin Elmer Optima 320 RL equipment (Waltham, MA, USA).

The metal contact surface with the solution is 100 mm^2^. The surfaces of the samples were observed using a SEM (JEOL JSM 5410 Microscopy, Tokyo, Japan) equipped with an x-ray microanalysis LZ5 EDS (Jeol, Tokyo, Japan) operated at 10 kV, which was also used for determining the chemical composition.

The data was statistically analyzed using Student’s t-tests, one-way ANOVA tables, and Turkey’s multiple comparison tests to evaluate any statistically significant differences between the samples with a *p*-value < 0.005.

## 3. Results

### 3.1. Stainless Steel

Figure 1 shows the surfaces at 1, 4, 7, and 14 days of immersion in the most concentrated mouthwash for the stainless steel archwires. The attack on the surface can be seen beginning on the fourth day and increasing as the immersion time increases. The orientation of the attack can be clearly seen following the direction of the wire lamination.

The results of ion release at different immersion times in mouthwashes with different sodium fluoride contents can be seen in Figure 2 for iron, chromium, and nickel. The iron ion release after 14 days in a 380 ppm fluoride solution reaches 7000 ppb, for Cr around 1000 ppb, and for nickel 500 ppb.

### 3.2. Ti-Mo

Figure 3 shows the surfaces at 1, 4, 7, and 14 days of immersion in the most concentrated mouthwash for the Ti-Mo archwires.

The fluoride-containing solution attacked the Ti-Mo surface aggressively. This is due to the ease of fluoride attack on titanium, as can be seen in Figure 4, where the release of titanium ions reaches 140,000 ppb. It can be observed that the dissolution of molybdenum is significantly lower (20,000 ppb). It can also be observed that the attack is not uniform on the surface, but rather localized “pitting” attacks occur on the crystalline grains with an orientation that favors the attack of the fluoride. This attack, which generates surface roughness, will cause a decrease in the ease of sliding the bracket on the Ti-Mo wire [25].

As in other studies, it has been observed that the aggressiveness of fluoride is not linear, with a sharp increase in ionic release when its concentration exceeds 300 ppm [35,36]. The titanium ion release after 14 days immersed in 130 and 200 ppm was 950 and 1250 ppb, respectively; for molybdenum, it was 290 and 380 ppb.

### 3.3. NiTi

Figure 5 shows surfaces at 1, 4, 7, and 14 days of immersion in the most concentrated mouthwash for the NiTi archwires.

In the case of NiTi, uniform surface etching is observed. No pitting or localized attacks can be seen. This is due to the fact that the two elements that make up the alloy are easily dissolved in the fluoride solution. The nickel depletion is higher than that of titanium but of the same order. The release of nickel and titanium ions can be seen in Figure 6.

As can be observed, the nickel release after 14 days immersed in 380 ppm produces an ionic release of around 210,000 ppb and for titanium, 180,000 ppb. The same behavior is seen when compared to TiMo archwires; the high concentration of fluorides causes a significant increase in ion release. At the same time, at 130 and 200 ppm, fluoride concentrations were around 850 and 9000 ppb for nickel and 1100 ppb and 6000 ppb for titanium.

It is observed that after 7 days of immersion in fluorides, the appearance of martensitic plates can be seen in the microstructure. This fact is produced by the great release of nickel and titanium ions. As more nickel ions are released than titanium ions, the chemical composition of the wire is enriched in titanium, resulting in higher transformation temperatures. The original NiTi wires are superelastic and cause dental movements. However, changes in the chemical composition cause Ms’s temperature to rise above 37 °C, and martensite, which has no superelasticity, is formed [35,37]. This martensite can be seen in Figure 6 for the wire after 7 days of fluoride immersion.

When the immersion time is increased, precipitates can be seen as early as day 10 and more clearly on day 14. These precipitates are rich in titanium since the decrease in nickel content reaches the stoichiometry of the Ti_2_Ni compound, which, in addition to losing its superelastic properties completely, causes great brittleness [35]. The precipitates can be seen in Figure 7.

## 4. Discussion

Mouthwashes based on sodium fluoride solutions are very effective preventive oral devices to avoid dental caries, and this is the reason these mouthwashes have been widely used in fixed orthodontics. Because these solutions change the pH to an acidic state, corrosion resistance decreases due to the breakdown of surface oxide protective films (Cr_2_O_3_ for austenitic stainless steel and TiO_2_ for TiMo and NiTi alloys) [38,39,40].

In austenitic steel, nickel is the primary austenite stabilizer element, and the atoms are formed in substitutional solid solution; in any case, Ni presents a high bond energy because the atoms do not form an intermetallic compound [41]. In the results shown in Figure 2, the slow ion release from the archwire can be observed. After 14 days of immersion in 380 ppm of fluorides, the Ni release is about 500 ppb. For chromium ions, the release was around 1000 ppb under the same conditions. Iron ions present values in greater quantities than chromium and nickel due to the higher content in steel. The chromium ions are not hexavalent ions that could cause cancer, but rather have valence 3. We can also see that the chromium release values are relatively small, much lower than those that occur in hip prostheses when the femoral ball is made of stainless steel. It has been described in traumatology that the wear of the femoral ball leads to ppm values, which are therefore very far from those produced by dissolution due to the effect of the mouthwash. In the case of traumatology, in addition to having an adverse physiological environment, the metal undergoes high stresses and friction forces that favor the chemical degradation of the metal [2].

According to the European Council Directive for the quality of water for human consumption, the maximum admissible for nickel ions is 20 μg/L and the average chromium levels in drinking water are 0.43 μg/L. Daily amounts of chromium and nickel intake from foods are 5 to 100 μg and 300 to 500 μg, respectively [42,43].

The ions released from stainless steel orthodontic wires in this study were insignificant when compared with the daily food and water intake. For the stainless steel the worst case-scenario was immersion in 380 ppm fluorides concentration for 14 days that corresponding to 50 μg/L nickel release for nickel, 71 μg/L for chromium and 500 μg/L for iron. These results would be obtained if the archwire was immersed in mouthwash containing 380 ppm fluoride for 24 h. When compared to the amount of food and water consumed daily, the ions released from orthodontic devices in this study were insignificant, with fluoride contents of 130 and 200 ppm.

Pits produced on the stainless steel surfaces by the fluorinated solution can be seen in the direction of the drawing, since this is where the greatest energy is stored, and these are the first corrosion points that cause the release of ions. This fact could be reduced if the wires, after being drawn, were subjected to an annealing heat treatment, which would cause the elimination of the internal energy that favors corrosion [42,43,44].

The titanium ion release is very important in the TiMo archwire because of the high concentration of fluorides. However, the ion release is insignificant when the fluoride concentrations are between 130 and 200 ppm. The main problem with these archwires is the roughness produced by the pitting, which increases the friction coefficient. Slipping against brackets will make it extremely difficult to avoid correct orthodontic therapy.

NiTi wires present a very dangerous release of nickel ions, exceeding 250,000 ppb after 14 days of immersion in a 380 ppm fluoride concentration mouthwash. As can be seen, these values exceed the nickel concentrations recommended by the health authorities, and therefore mouthwashes with high fluoride concentrations should not be recommended by clinicians. The release of nickel ions is even higher than that of titanium ions, which is in the range of 180,000 ppb. Such a large release causes variations in the chemical compositions, increasing the martensitic transformation temperatures (M_s_ and M_f_), and stabilizing the martensitic phase at 37 °C, as was shown by SEM images (Figure 8). These martensite plates inhibit superelasticity and make the wire a wire that does not exert corrective stress and therefore inhibits its function. As is well known, increasing the martensitic transformation temperature reduces the transformation stresses, and thus the corrective stresses will decrease until martensite appears on the surface. At immersion times of 7 days or more, globular nickel-rich precipitates have been observed. These precipitates correspond to TiNi_2_ as can be observed in Figure 7, producing a great brittleness in the wires. The treatment of NiTi wires with 130 and 200 ppm mouthwashes shows values that, although high, do not exceed the recommendations of the health authorities [35]. It is observed that from 4 to 7 days of permanence in 200 ppm solutions, a significant change in ion release occurs, which could be explained by the onset of phase transformations due to changes in the chemical composition of NiTi. The austenite and martensite phases have different ion releases due to the change in atomic ordering. As in the case of TiMo wires, no cytotoxicity of titanium ions in the human body has been reported, although some studies have shown preferential accumulations in the kidney in experiments on rats [12], and it has not been possible to confirm the effects of titanium or the thresholds of danger of this element.

In the case of NiTi wires, as in the case of TiMo wires, effects on the topography of the wire are not observed. In this case, as the elimination is very high in both Ni and Ti elements, uniform defects are produced on the NiTi surface, not affecting the roughness, although, as we have seen, they do affect the microstructure of the wire.

From the results, it can be demonstrated that stainless steel arches are more stable than those of NiTi and TiMo when immersed in fluorinated mouthwashes. Their surfaces do not undergo significant erosion, as can be observed in Figure 3. Because nickel is the most common element that causes metal ion-induced contact allergy leading to dermatitis in humans, the worst case scenario was NiTi treated at high fluoride concentrations. Chromium is the second most common metal to cause allergic reactions, but the release in stainless steel is low. There is scientific evidence that Ni ions are carcinogenic, mutagenic, and produce cytotoxicity in cells [42,43,44]. Even a small amount of release might produce sensitivity when the orthodontic appliance is in place for 2 to 3 years. But, for an allergic reaction in the oral mucosa, an antigen must be 5–12 times greater than that needed for a skin allergy. Clinicians should be aware that metal ion release can cause a local hypersensivity reaction to nickel or chromium.

A limitation of this study is that metal is released into the oral cavity with saliva as the medium, and this could be influenced by a high chloride mixture from the intake of various foods and drinks with a low pH. Also, the characteristics of saliva change according to the patient’s health and the time of day. In this research, mouthwashes have been used in static conditions, but more metallic ion release could occur in real life because of the fluidity of saliva in the mouth and also due to the removal of the oxide layers by tooth brushing. Kerosuo et al. found a great deal of relief after using an oral functioning simulator apparatus to simulate the dynamic conditions of the mouth [45]. In addition, future studies should analyze the role of bone cements used for bracket fixation, calcium phosphates, and the use of antibacterial molecules [46,47,48].

Further research is needed to determine whether fluoride solutions on archwires cause dangerous effects in the oral cavity, possibly with long-term systemic consequences. It is known that nontoxic levels of metals like nickel can cause changes in DNA or inhibit DNA-restoring enzymes [42,45], which could have an adverse biological effect in the long term. A long-term evaluation should be conducted.

As we have seen, the differences in the chemical composition of NiTi are of great importance for its superelastic properties and transformation temperatures. The release of preferential Ni ions will cause variations in the M_s_ temperatures and therefore in the corrective stresses that the orthodontic wire makes on the tooth. The stresses decrease until the superelastic effect is lost and precipitates are formed, which make the superelastic wire lack corrective properties and embrittle the structure [49,50]. It is important to determine how variations in chemical composition affect the phases present in the orthodontic metals, as they can vary very significantly in properties [51,52,53].

A distinction must be made between two types of chemical degradation of orthodontic materials, namely the release of ions into the physiological environment and electrochemical corrosion. These are two different processes: a metallic material in contact with a medium releases metallic ions due to its solubility product until it reaches equilibrium, and corrosion requires a chemical reaction of oxidation-reduction in which a corrosion product is produced that is often toxic. It is important to distinguish the two processes, which are often linked but have different natures [54,55]. In the cases we have studied in this work, chemical reactions are not observed, but only the release of ions. Corrosion products generally cause black metallic oxides that are due to the oxidation of particles, generating a disease called metallosis. The metals must be removed, and the contaminated tissues must be cleaned. The release of ions goes to the physiological environment, and different studies have observed that they can be stored in different organs. The threshold concentrations that can cause toxicity in humans are not known, and at the moment, the quantities of ions are compared to those we ingest in our daily diet [56,57,58].

## 5. Conclusions

In this study, the release of metal ions was observed in the orthodontic wires studied when they were immersed in fluoride solutions. Ions released are significantly higher in high-concentration mouthwash treatments (380 ppm) than in lower-concentration treatments (130 and 200 ppm). The most stable wires are those made of stainless steel, where the release of nickel and chromium ions is within acceptable health-care parameters. However, the release of titanium ions in treatments with mouthwashes containing fluoride concentrations of 380 ppm results in a very high release of titanium ions, which causes pitting on the surface and will undoubtedly affect the wire’s good release from the bracket. The release of nickel and titanium ions in NiTi wires at high concentrations not only causes a release of nickel ions that can cause toxicity but also changes the microstructure, inhibiting the superelasticity of the orthodontic wire and its co-straightening function. Consequently, this work advises against recommending the use of mouthwashes with concentrations higher than 200 ppm when orthodontic wires are used.

## Figures and Tables

**Figure 1 ijerph-20-02780-f001:**
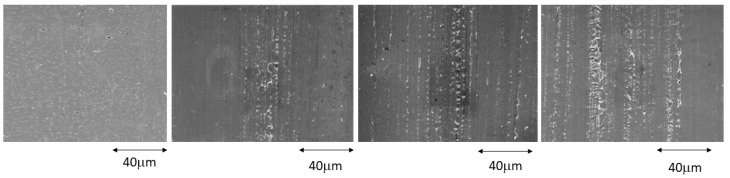
Stainless steel surfaces after 1, 4, 7, and 14 days immersed in a 380 ppm fluoride solution.

**Figure 2 ijerph-20-02780-f002:**
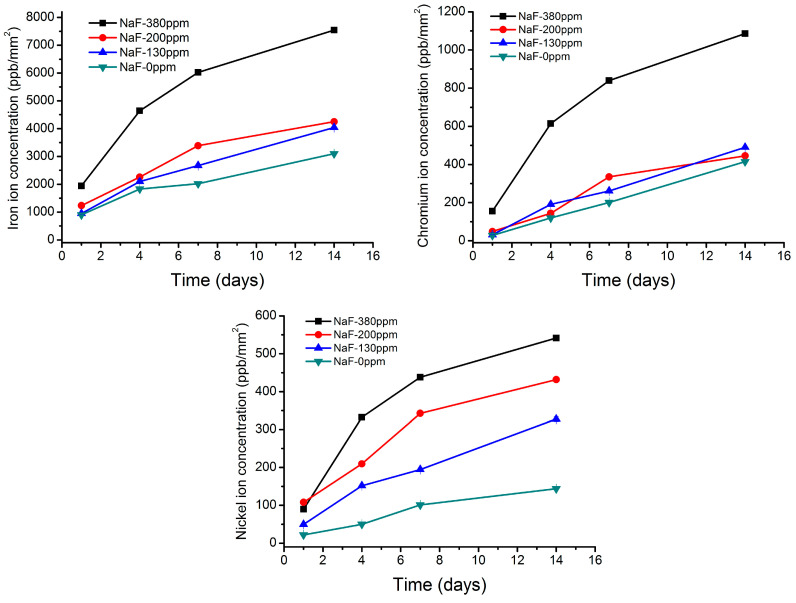
Iron, chromium, and nickel ions release at different times of immersion and in different fluoride concentration solutions. The contact surface of the archwire with the solution is 100 mm^2^.

**Figure 3 ijerph-20-02780-f003:**
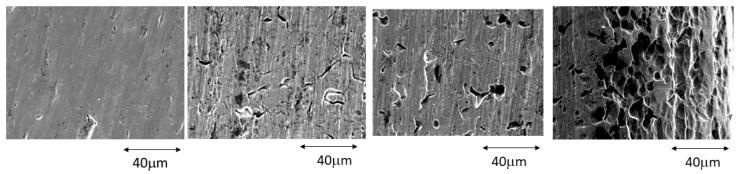
Ti-Mo surfaces after 1, 4, 7, and 14 days immersed in a 380 ppm fluoride solution.

**Figure 4 ijerph-20-02780-f004:**
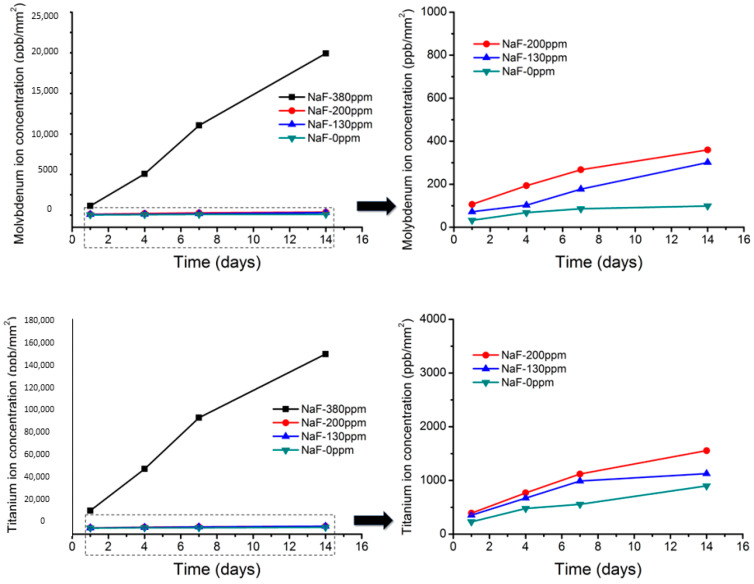
Titanium and molybdenum ion release at different times of immersion and at different fluoride concentrations solutions. At concentrations of 130 and 200 ppm, the figure has been magnified. The contact surface of the archwire with the solution is 100 mm^2^.

**Figure 5 ijerph-20-02780-f005:**
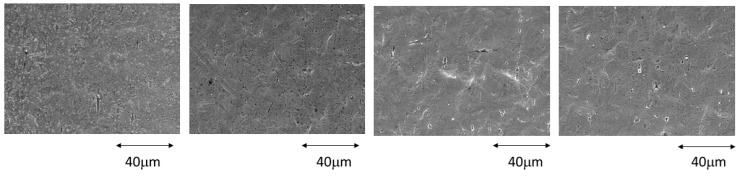
NiTi surfaces after 1, 4, 7, and 14 days immersed in a 380 ppm fluoride solution.

**Figure 6 ijerph-20-02780-f006:**
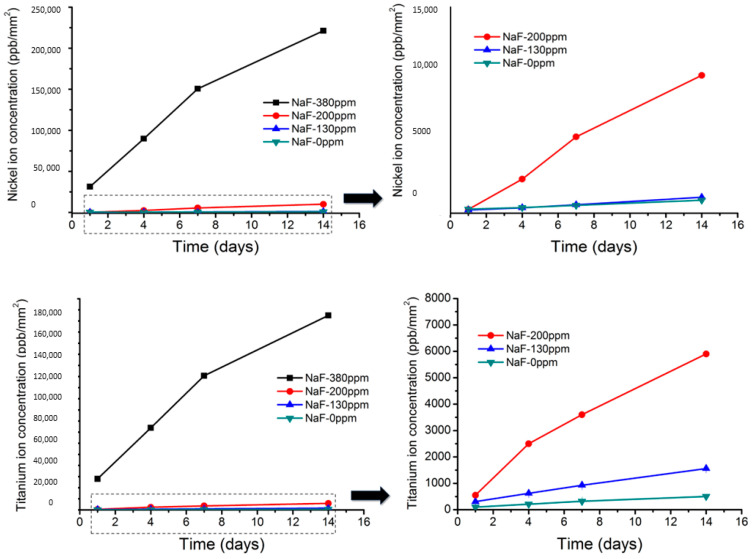
Nickel and titanium ion release at different times of immersion and at different fluoride concentrations solutions. At concentrations of 130 and 200 ppm, the figure has been magnified. The contact surface of the archwire with the solution is 100 mm^2^.

**Figure 7 ijerph-20-02780-f007:**
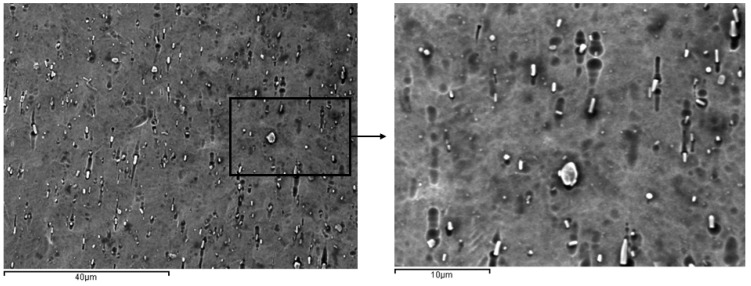
NiTi archwire surface after the immersion for 14 days in mouthwashes with 380 ppb of fluoride. Titanium-rich precipitates are observed on the surface.

**Figure 8 ijerph-20-02780-f008:**
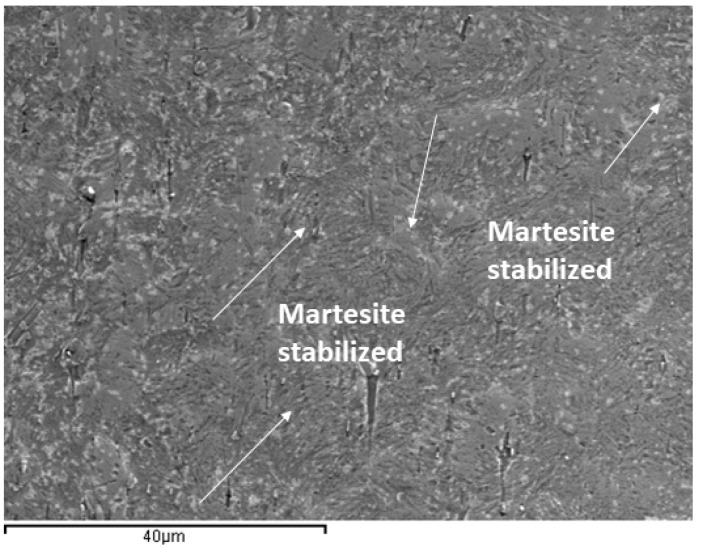
NiTi archwire surface after 7 days of immersion in mouthwashes with 380 ppb fluorides. Martensitic plates are observed.

**Table 1 ijerph-20-02780-t001:** Chemical compositions of the orthodontic archwires studied (% of weight).

Materials	Brand	Ni	Ti	Mo	Cr	Fe	C
Stainless steel	American Orthodontics. Sheboygan, WI, USA	14.8		3.0	18.0	64.2	0.02
Ti-Mo	Beta Blue. Highland Metals, Bangkok, Thailand		87.0	13.0			
Ni-Ti	Neo Sentalloy. GAC, West Columbia, USA	55.8	44.2				

**Table 2 ijerph-20-02780-t002:** NaF composition of the different mouthwashes.

Mouthwahes	NaF (ppm)
0	0
1	130
2	200
3	380

## Data Availability

The authors can provide details of the research requested by letter and comment on their needs.

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
