# Peer review of "Effect of Fluoride Content of Mouthwashes on the Metallic Ion Release in Different Orthodontics Archwires"

_ijerph, 2023, doi:10.3390/ijerph20042780_

Round 1

Reviewer 1 Report (Previous Reviewer 3)

In this manuscript, three commercialized archwires of different alloys were tested in mouthwashes at 3 different fluoride concentrations and the tests were lasted for different times. The metallic ion release conditions were measured by using ICP-MS and the surface of the materials were checked by using SEM. Overall, this manuscript was well-written with good organization, fairly reasonable experimental design and analysis, and the conclusion is based on the experiments and analysis. The manuscript is also well-referenced, although more recent papers could be referenced.  

There are several questions/problems:

1.      In figure 1,3,5, there is a double arrowed line and 40µm beside it. Is that the scale?

2.      In two graphs of Figure 4 and Figure 6, the concentrations do not include 300 ppm. Why?

3.      Quite a lot grammar errors, such as line: 52, 69, 71, 98, 100, 102, 108, 114, 177, 185, 207, 242, 272, and 276

Author Response

REVIEWER 1

Dear Reviewer,

Thanks for taking the time to review our manuscript and suggest to us to improve our work by providing a lot more detail. We have done so, and we are now submitting a manuscript that not only addresses the points you specifically raised but also many others that we have considered in order to deliver what we think is a much-improved version of our work. This version includes more paragraphs, figure, English grammar revisions in all main sections, new references. Thanks a lot. We are looking forward to your comments.

Sincerely,

Francisco-Javier Gil Mur

In this manuscript, three commercialized archwires of different alloys were tested in mouthwashes at 3 different fluoride concentrations and the tests were lasted for different times. The metallic ion release conditions were measured by using ICP-MS and the surface of the materials were checked by using SEM. Overall, this manuscript was well-written with good organization, fairly reasonable experimental design and analysis, and the conclusion is based on the experiments and analysis. The manuscript is also well-referenced, although more recent papers could be referenced.  

There are several questions/problems:

  1. In figure 1,3,5, there is a double arrowed line and 40µm beside it. Is that the scale?

Yes, this is scale bar for all microstructures.

  1. In two graphs of Figure 4 and Figure 6, the concentrations do not include 300 ppm. Why?

The figures are an enlargement of the ion release values for 0, 180 and 200 ppm of sodium fluoride. The differences in the first figure are very small due to the high ion release values of the mouthwashes with 380 ppm flouride. With this figure in more detail the differences can be seen.

  1. Quite a lot grammar errors, such as line: 52, 69, 71, 98, 100, 102, 108, 114, 177, 185, 207, 242, 272, and 276.

All errors commented by the reviewer have been corrected. Thank you very much

Reviewer 2 Report (New Reviewer)

The manuscript describes the investigation of the metal ion release in austenitic stainless steel, Ti-Mo and superelastic NiTi archwires immersed in three mouthwashes with different fluoride concentrations.

The topic is not novel, the scope is narrow; the methods are limited and results are appropriate for achieving the objectives. A novelty statement is needed.

However, the manuscript is not well prepared, in that the submitted version still has tracked changes.

English usage needs to be improved. The reproducibility is relatively low with the details of the main experiment Ion-release quantification missing.

The manuscript misses introducing the crystal structures of these alloys, for example, SS’s body-centered cubic α’ martensite accompanying the face-centered cubic γ austenite, etc. These may help the readers understand better the structural differences.

In conclusion, a major revision is needed before further consideration in Int. J. Environ. Res. Public Health.

More details for consideration:

1. When listing terms, please use “and” rather than “,” between the last and second last terms. This applies to the first two sentences in Introduction.

2. Some problematic English usage should be corrected, for example but not limited to: … should be not recommended…; … has been hypersensitive reaction to this metal; During the orthodontic treatment can…; This fact leads the use…; … such friction coefficients…; … and the bracket avoiding the teeth movement…

3. How was the 100 mm2 contact surface with the solution determined?

4. The statistical analysis, which confidence level was applied, what is alfa value?

5. Fig. 6 has different style from the previous ones somehow. Please use the same style.

6. Markings and arrows in figures are not easy to see, change font color to white.

7. fluorides -> fluoride

8. Mouthwashes are not devices but treatment; not to avoid dental caries but to prevent them.

9. It is recommended that the authors also cite some recent literature that elucidated the relationship between composition, microstructure and surface characteristics of archwires of the 3 alloy types and performance, especially corrosion resistance: stainless steel doi:10.4012/dmj.2016-206, 10.3390/nano9081119, 10.1039/C7JA00065K; Ti-Mo and NiTi 10.3390/met12030406

Author Response

REVIEWER 2

Dear Reviewer,

Thanks for taking the time to review our manuscript and suggest to us to improve our work by providing a lot more detail. We have done so, and we are now submitting a manuscript that not only addresses the points you specifically raised but also many others that we have considered in order to deliver what we think is a much-improved version of our work. This version includes more paragraphs, figure, English grammar revisions in all main sections, new references. Thanks a lot. We are looking forward to your comments.

Sincerely,

Francisco-Javier Gil Mur

The manuscript describes the investigation of the metal ion release in austenitic stainless steel, Ti-Mo and superelastic NiTi archwires immersed in three mouthwashes with different fluoride concentrations.

The topic is not novel, the scope is narrow; the methods are limited and results are appropriate for achieving the objectives. A novelty statement is needed.

However, the manuscript is not well prepared, in that the submitted version still has tracked changes.

English usage needs to be improved. The reproducibility is relatively low with the details of the main experiment Ion-release quantification missing.

English has been revised. Many mistakes and sentences have been improved.

The manuscript misses introducing the crystal structures of these alloys, for example, SS’s body-centered cubic α’ martensite accompanying the face-centered cubic γ austenite, etc. These may help the readers understand better the structural differences.

These crystal structures have been introduced in the Introduction according to the reviewer.

In conclusion, a major revision is needed before further consideration in Int. J. Environ. Res. Public Health.

More details for consideration:

  1. When listing terms, please use “and” rather than “,” between the last and second last terms. This applies to the first two sentences in Introduction.

Done

  1. Some problematic English usage should be corrected, for example but not limited to: … should be not recommended…; … has been hypersensitive reaction to this metal; During the orthodontic treatment can…; This fact leads the use…; … such friction coefficients…; … and the bracket avoiding the teeth movement…

These sentences have been improved.

  1. How was the 100 mm2contact surface with the solution determined?

The wires are cylindrical and were cut with a high precision cutting machine to obtain this wire surface.

  1. The statistical analysis, which confidence level was applied, what is alfa value?

This parameter has been introduced in line 112.

  1. 6 has different style from the previous ones somehow. Please use the same style.

Done

  1. Markings and arrows in figures are not easy to see, change font color to white.

Done

  1. fluorides -> fluoride

Done

  1. Mouthwashes are not devices but treatment; not to avoid dental caries but to prevent them.

The preventive nature of mouthwashes has been added in different sentences of the text.

  1. It is recommended that the authors also cite some recent literature that elucidated the relationship between composition, microstructure and surface characteristics of archwires of the 3 alloy types and performance, especially corrosion resistance: stainless steel doi:10.4012/dmj.2016-206, 10.3390/nano9081119, 10.1039/C7JA00065K; Ti-Mo and NiTi 10.3390/met12030406

The references have been introduced in the discussion according to the reviewer. We have added some considerations about the importance of the chemical composition and microstructure on the properties.

Reviewer 3 Report (New Reviewer)

The manuscript needs major revision.

Introduction: Please mention the novelty of your study

 Materials & Methods: Pleases mention sample size calculation

Results: It would be better that ion release reported per unit area

Results: The standard deviation of ion concentration should be added.

Discussion:  In the discussion section I would like to see a more profound discussion about the findings. What is the meaning of your results in light of earlier studies?

References: You could increase the number of more recently studies in the introduction and discussion sections

Author Response

REVIEWER 3

Dear Reviewer,

Thanks for taking the time to review our manuscript and suggest to us to improve our work by providing a lot more detail. We have done so, and we are now submitting a manuscript that not only addresses the points you specifically raised but also many others that we have considered in order to deliver what we think is a much-improved version of our work. This version includes more paragraphs, figure, English grammar revisions in all main sections, new references. Thanks a lot. We are looking forward to your comments.

Sincerely,

Francisco-Javier Gil Mur

Introduction: Please mention the novelty of your study

The novelty has been introduced in the last paragraph of the Introduction according to the comment of the reviewer

Materials & Methods: Pleases mention sample size calculation

The calculation of the samples has been introduced in Materials & Methods.

Results: It would be better that ion release reported per unit area

Done

Results: The standard deviation of ion concentration should be added.

The standard deviation is incorporated in the graphs. However, the ICP-MS equipment has a sensitivity of 10 ppb and the different ion release assays performed are very reproducible. The error bars are smaller than the symbols used to mark the results.

Discussion:  In the discussion section I would like to see a more profound discussion about the findings. What is the meaning of your results in light of earlier studies?

Several paragraphs have been incorporated in the text, in the results and the discussion according to the comment of the reviewer. Thank you

References: You could increase the number of more recently studies in the introduction and discussion sections

New recent references have been introduced in the discussion

Round 2

Reviewer 2 Report (New Reviewer)

The main issues have been addressed and the overall quality has been improved. I recommend publication after a few minor issues are addressed:

1. Fig, 1,3,5, each SEM micrograph needs its own scale bar.

2. p-value<0.005? 0.5%? Is this correct?

Author Response

Thank you again for your help.

Figures 1, 3 and 5 have its own scale bar for each image.

Yes p<0.005 is 0.5% or p>99.5%.

Reviewer 3 Report (New Reviewer)

The manuscript has been improved and it an be accepted.

Author Response

Thank you very much for your help

This manuscript is a resubmission of an earlier submission. The following is a list of the peer review reports and author responses from that submission.

Round 1

Reviewer 1 Report

The scale of Figure 1 is inconsistent.

The concertration of metal ion release should be converted into unit area, which is more convincing.

In Figure 6, why does the 4-7d data show a curve?

The style of the charts was not uniform throughout the manuscript.

Reviewer 2 Report

Generally, the topic of this study is interesting and it has a good clinical significance. Some concerns are listed as follow:

1. In this study, the metal wire was immersed in the mouthwash for 1-14 days. However, in our real life, the mouthwash stays in the mouth for about 1-2 min at most. How to interpret this difference?

2. the clinical significance should be stated in more detail and the clinical guide for the dentist or patient should be summarized based on the conclusion of this study. 

Reviewer 3 Report

The effect of fluoride content of mouthwashes at four different concentrations on the metallic ion release in three different orthodontics archwires were investigated. ICP-MS was used to determine the concentration of ions released and SEM was used to observe the surface changes of the archwires. The manuscript has clear introduction, fairly good experimental design, as well as discussion (I like the discussion part) and conclusion. Studies were well referenced.

Problems:

1.      1. The language of this manuscript needs to be improved. Grammar errors:

At least lines: 33, 39, 42, 43, 52, 68, 74, 82, 94, 106, 140, 190, 191, 197, 204, 214, 216, 217, 253, 266.

2.    2. What is the reason to choose the three different fluoride concentration: 130, 200 and 380 ppm? More concentrations should be included. Especially, some results show abrupt change at 380 ppm. At least one concentration between 200 and 380 ppm should be considered.

3.     3.  Line 61: in physiological media form HF because all sodium salts are soluble in aqueous media. Please provide more supporting information about this statement.

4.      4. The scales of SEM images should be larger.